# Phenolic Compounds’ Occurrence in *Opuntia* Species and Their Role in the Inflammatory Process: A Review

**DOI:** 10.3390/molecules27154763

**Published:** 2022-07-25

**Authors:** Walid Zeghbib, Fares Boudjouan, Vitor Vasconcelos, Graciliana Lopes

**Affiliations:** 1Laboratoire de Biochimie Appliquée, Faculté des Sciences de la Nature et de la Vie, Université de Bejaia, Bejaia 06000, Algeria; walidzeghbib1993@gmail.com; 2CIIMAR—Interdisciplinary Centre of Marine and Environmental Research, Terminal de Cruzeiros do Porto de Leixões, Avenida General Norton de Matos s/n, 4450-208 Matosinhos, Portugal; vmvascon@fc.up.pt; 3Laboratoire de Génie de l’Environnement, Faculté de Technologie, Université de Bejaia, Bejaia 06000, Algeria; fares0501@gmail.com; 4Faculté des Sciences de la Nature et de la Vie, Université de Bejaia, Bejaia 06000, Algeria; 5FCUP—Faculty of Sciences, University of Porto, Rua do Campo Alegre s/n, 4169-007 Porto, Portugal

**Keywords:** *Opuntia* sp., prickly pear, phenolic compounds, flavonoids, inflammation

## Abstract

Within the *Cactaceae* family, *Opuntia* comprises the most widespread species, with a recognized importance in human life, including feeding, domestic use as home natural barriers, and as a traditional remedy for diverse diseases and conditions such asthma, edema, and burns. Indeed, scientific reports have stated that these health benefits may be due to various active compounds, particularly polyphenols, which are ubiquitously found in plants and have proven their pharmacological efficiency by displaying antimicrobial, anti-cancer, and anti-inflammatory activities, among others. *Opuntia* species contain different classes of phenolic compounds that are recognized for their anti-inflammatory potential. Among them, quercetin, isorhamnetin, and kaempferol derivates were reported to greatly contribute to modulate cells’ infiltration and secretion of soluble inflammatory mediators, with key implications in the inflammatory process. In this review, we make a summary of the different classes of phenolic compounds reported in *Opuntia* species so far and explore their implications in the inflammatory process, reported by in vitro and in vivo bioassays, supporting the use of cactus in folk medicine and valorizing them from the socio-economic point of view.

## 1. Introduction

Cacti belong to the plant family *Cactaceae*, which includes around 1500 species with recognized and very high phenotypic variations. Although the taxonomical characterization of the species is not always easy [1,2], the main subfamilies are currently well classified (Figure 1). Among them, the genus *Opuntia*, from the *Opuntioideae* subfamily, is the most recognized, being widely distributed across the globe [3,4]. *Opuntia* cacti (prickly pear) are among the plants with greater recognition in quotidian life, with archeological evidence encompassing the analysis of human coprolites dating back between 6500 to 10,000 years ago and suggesting its use as foodstuff [5,6]. The first discovery of cacti dates back to the times of the conquest of the new world by Spanish conquistadors who, besides being amazed by its attractive and delicious fruits, also noticed its granted economic and cultural importance in the daily life of the ancient Mesoamerican population [7,8] (Figure 2). After this, the plant became widely spread worldwide through cultivation and trade, nowadays being abundant in many arid and semi-arid regions of America, Africa, Asia, Europe, and Oceania [9,10].

The fast growth of cactus species and their good adaptation to poor soil make them very important plants for these populations, being part of their nourishment, livestock feed, and as a home natural barrier [11,12]. In addition, cacti became important in folk medicine for their capacity to alleviate diverse health conditions, such as diarrhea, asthma, hemorrhoids, ulcers, burns, and edema [2,13,14]. These beneficial properties to human health led to an increase in scientific research focused in cacti plants, particularly on species of the genus *Opuntia*, due to their recognized pharmacological properties [6]. Indeed, it was revealed that all parts of cacti (flowers, fruits, cladodes, and peels) constitute undeniable sources of valuable nutritional elements and biologically active primary and secondary metabolites, such as vitamins, carotenoids, betalains, polyunsaturated fatty acids, and polyphenols, leading people to consider them as important functional foods, with interesting nutraceutical and pharmacological properties [15,16,17]. 

Phenolic compounds, including flavonoids and phenolic acids, are ubiquitous molecules found in nature, particularly in plants, with more than 8000 compounds described so far, and divided into different classes [18]. Among other factors, the phenolics’ qualitative and quantitative profiles vary with plants’ genus, species, ripeness, cultivar, growth region, and kind of plant tissue [19,20,21]. The literature has reported a multitude of phenolic compounds in all *Opuntia* species [6,22], with a particular prevalence of phenolic acids and flavonoids, such as dihydroquercetin, quercetin, isorhamnetin, and kaempferol, known for their efficient antioxidant activity and ability to protect human organisms from the deleterious effects of free radicals through diverse mechanisms of action. It is widely known that oxidative stress appears as a consequence of tilting the balance in favor of free radicals compared to the antioxidant system. This imbalance stands at the base of many diseases, including different types of cancer, arteriosclerosis, myocardial infarction, diabetes, inflammatory diseases, central nervous system disorders, and cells’ aging [23,24]. In addition to the antioxidant power of polyphenols, recent investigations also recognized their antimicrobial, hepatoprotective, anti-carcinogenic, and anti-inflammatory properties [22,25]. As a matter of fact, research devoted to phenolic compounds’ valorization is ever present in the majority of natural matrices, from the optimization of the extraction processes to their biotechnological exploitation in food and cosmetic industries and elucidation of mechanisms of action in a wide array of pharmacological targets. Following the traditional use of cacti in acute health conditions where inflammation plays a central role, inflammatory mediators and enzymes appear among the main targets. Inflammation is a body’s natural response to a pathogen invasion, toxin, or physical damage (chemical or traumatic), which involves the generation of a wide array of inflammatory mediators, such as reactive oxygen species (ROS), by inflammatory and immune cells. When the inflammatory process is uncontrolled or when the endogenous defense systems fail to establish homeostasis, inflammation can become chronic, leading to tissue damage and often preceding the establishment of chronic diseases [26,27]. There are many pharmacological treatments for inflammation based on steroidal and non-steroidal compounds; however, they present significant undesirable side effects and resistances, leading to an increasing interest in the search for bioactive compounds from natural sources as a potential effective and alternative non-pharmacological approach [28]. In this regard, the present work provides a general review on the different classes of phenolic compounds found in *Opuntia* sp. and on the anti-inflammatory activity reported for the genus so far.

## 2. Phenolic Compounds

### 2.1. General Overview

After cellulose, phenolic compounds represent the most abundant group of secondary metabolites of the plant kingdom. This large family ranges from simple compounds with low molecular weight to large and complex polyphenols mainly found conjugated with sugars and organic acids [29]. In plants, phenolic compounds are biosynthesized by the shikimate pathway, which is localized in the chloroplasts. These aromatic molecules have important roles in plants, being implicated in the regulation of their growth, signaling, defense, and in conferring color to their fruits, leaves, and flowers [24,30,31]. Their chemical structure is characterized by the presence of at least one aromatic ring containing one or more hydroxyl groups [32]. From the chemical point of view, phenolic compounds are characterized by an acidic behavior, since the oxygen of the hydroxyl function is strongly linked to the ring, while the connection to the hydrogen atom is weak, allowing the proton dissociation into the medium and giving origin to a negatively charged phenolate ion [33].

The medicinal properties reported for polyphenols over the years aroused scientists’ interest in improving their extraction methodologies by using different solvents and extraction methods, the most widely explored being infusion, decoction, and maceration as well as Soxhlet, ultrasound, and microwave-assisted extraction [34]. There are more than 8000 phenolic compounds described in plants, with a high structural variability [18,35]. According to the classification system followed by De la Rosa et al. [30], phenolic compounds can be divided according to their chemical structure into two main classes: flavonoids and non-flavonoids. The first category is characterized by its structure complexity and known for its efficient bioactivity, accounting for nearly two-thirds of dietary polyphenols [30]. Its basic structure consists of a 15-carbon structure with two phenyl rings (A and B) connected by a three-carbon bridge, forming a heterocyclic pyran ring (ring C) skeleton (Figure 3a). The differences in the pyran ring substituents and the extent of hydrogenation allows defining six subcategories: flavones, flavonols, flavanols, isoflavones, flavanones, and anthocyanidins [29,31].

The non-flavonoids’ category includes smaller and simpler compounds. The principal molecules in this category are phenolic acids, particularly present in fruits and vegetables, accounting for one-third of dietary phenolic compounds. Phenolic acids are structurally composed by a single benzene unit, substituted by one carboxylic group and at least one hydroxyl group. Thus, many compounds can be considered simple phenols, and they are generally classified according to the number of carbons they have, the most common being hydroxybenzoic acids with a basic skeleton C_6_–C_1_ (Figure 3b) [30,33]. However, the non-flavonoids’ category also includes other compounds with a complex structure and with a high molecular weight, which are characteristic and major components of some plants, such as lignans, chalcones, and stilbenes [30].

Furthermore, the significant differences between species reported in the literature make it difficult to establish the qualitative and quantitative phenolic profiles of *Opuntia* sp. These differences are mostly devoted to the different extraction methods followed by the authors, which include both the use of solvents of different polarities, different extraction times, temperatures, and equipment. Differences in the *Opuntia* raw material used for extraction are also worth considering: phenolics’ extractions can be performed using dry or fresh material, of different maturation states, and from different geographic locations, which modifies the abiotic factors to which species are exposed and, consequently, their phenolic profile. For instance, Chahdoura et al. [60] found that ferulic acid derivatives were the most abundant compounds in *Opuntia* sp. seeds, reaching about 0.36 and 0.95 mg/g for *Opuntia microdasys* (Lehm.) N.E. Pfeiffer and *Opuntia macrorhiza* Engelm., respectively, while Amrane-Abider et al. [46] found a higher chlorogenic acid content, with 0.89 mg/g in *Opuntia ficus-indica* L. (Mill) seeds. The most abundant phenolic acid from the pulp of different varieties of *O. ficus-indica* found by the authors was piscidic acid, with 8.70–22.31 mg/g, and quercetin was the most abundant flavonoid with 0.08–0.26 mg/g (dry weight, DW). Contrarily, Zenteno-Ramírez et al. [17] reported gallic acid and epicatechin as the most abundant compounds in the pulp of different *Opuntia* species. In the peel of different *O. ficus-indica* varieties, García-Cayuela et al. [57] found that piscidic acid was the most abundant phenolic acid with 27.53–44.62 mg/g DW and isorhamnetin derivates were the most representative flavonoids (1.48–2.54 mg/g DW). For the cladodes, some reports indicated that quinic acid and myricetin were the most common in *Opuntia dillenii* (Ker Gawl.) Haw. cladodes [39,56], while Missaoui et al. [41] reported for *O. ficus-indica* a higher abundance for piscidic acid and isorhamnetin derivates, with 9.67 mg/g and 3.93 mg/g, respectively. Regarding the flowers, a study by Chahdoura et al. [59] on the flowering stage of *O. microdasys* demonstrated that ferulic acid and isorhamnetin derivates were the most frequent, ranging from 1.24–2.95 mg/g and 4.68–23.04 mg/g, respectively, while Ammar et al. [65] found a higher content in quinic acid and quercetin derivates, with 1.32 and 8.50 mg/g, respectively, for *O. ficus-indica* flowers. Moreover, Ouerghemmi et al. [54] reported that ferulic acid and quercetin were in higher amounts when compared to other phenolic compounds in *O. ficus-indica* flowers. It seems evident, based on the available studies, that besides the abiotic factors, the species-specific ones have a significant role in phenolics’ concentration and distribution throughout the different plant tissues. It seems difficult to establish a tissue fingerprint for *Opuntia* sp. since both flavonoids and phenolic acids present a wide distribution throughout all the studied plant parts. Based on the available studies, it seems that seeds present the widest variety of phenolic compounds, while the lowest variability has been observed for flowers. It was also observed that anthocyanidins and hydroxycinnamic acid were almost exclusive of cladodes; however, the fact that this was reported in only one study is not enough to state it as a tissue fingerprint or a species-specific characteristic. The same line of thought can be followed for phloretin, psoralen, and pinoresinol, which were only reported in the seeds of *Opuntia stricta* (Haw.) Haw.

### 2.2. Phenolic Compounds’ Occurrence in Opuntia sp.

Different classes of phenolic compounds can be found in cactus plants, with different qualitative and quantitative profiles, which mainly depends on environmental factors, plant origin, species, developmental stage, and age [2]. From the literature, there are no records about a specific extraction method or the most suitable solvent for polyphenols’ recovery from *Opuntia* sp. samples; hence, researchers tend to choose the procedure that suits them according to their objectives and equipment availability. Accordingly, due to the polarity of polyphenols, some high polar solvents have been generally used, such as methanol, acetone, ethanol, water, or even mixtures between them [36,37]; several extraction methods have been followed, including conventional ones, such as maceration, infusion, and decoction [10,38,39], and non-conventional ones, such as microwave-assisted extraction and ultrasonic-assisted extraction [40,41].

The occurrence of phenolic compounds in cactus has been described in seeds, flowers, cladodes, pulp, and peel, as detailed in Table 1. According to Santos-Díaz et al. [6], the phenolic profile in *Opuntia* genera is complex, with more than 40 compounds described in both pulp and cladodes and more than 20 in seeds of different species. It seems important to highlight that the majority of the available reports do not specify the phenolic composition of each vegetative tissue of *Opuntia* sp., which makes it difficult to establish a defined qualitative profile. However, some compounds such as delphinidin, petunidin, and malvidin seem to be exclusively found in *Opuntia* sp. cladodes, while phloretin, psoralen, pinoresinol, and epigallocatechin are found in seeds.

## 3. *Opuntia* sp. in Inflammation

Inflammation is a physiological, self-limiting process occurring in mammalian tissues as a response to harmful situations such as microorganism invasion, physical damage, or exposition to toxic chemicals. The inflammatory process tends to eliminate primary triggers and contributes to initiating the regeneration of injured tissues by mediating an organized immune response, involving particularly macrophages and mast cells [66,67]. However, in some situations, the mechanisms involved in restoring tissues’ homeostasis fail, generating a deregulated response that often results in a chronic inflammatory response, which is ever present in a wide variety of diseases and metabolic disorders such as diabetes, obesity, cancer, arthritis, and neurodegenerative and cardiovascular diseases [66,68].

The inflammatory framework involves a complex cascade of events with a coordinated action between pro- and anti-inflammatory mediators and biological systems including different cell lines (macrophages, neutrophils) and signaling molecules [69,70]. The NF-κB transcription factors have been long recognized for constituting a prototypical pro-inflammatory signaling pathway. In fact, these proteins are normally retained in the cytoplasm, being bound to a class of inhibitory proteins known as the IκB family. However, after stimulation, the activation of specific enzymes, known as IκB kinases (IKK), may phosphorylate the inhibitory protein, leading to the dissociation of the IκB/NF-κB complex. This results in a proteasomal degradation of IκB protein, while NF-κB can then translocate to the nucleus, binding DNA and activating the transcription of some targeted genes for cytokines, such as tumor necrosis factor-α (TNF-α) and interleukins (IL) (IL-1β and IL-6), as well as the production of several enzymes including cyclooxygenase (COX) and lipoxygenase (LOX) (Figure 4) [71,72,73]. These latter enzymes have a key role in the inflammatory process through the transformation of the arachidonic acid released from the phospholipid membrane into a spectrum of pro-inflammatory bioactive mediators including prostanoids and leukotrienes, which act by enhancing edema formation, increasing vascular permeability and leukocytes’ infiltration into the injured tissue [74,75,76].

Over the years, many studies have been conducted in order to explore the potential targets of *Opuntia* sp. extracts and isolated phenols regarding the inflammatory process. From the anti-inflammatory investigations of *Opuntia* sp. reported in the literature, some species were the subject of greater scrutiny: *O. stricta*, *Opuntia humifusa* (Raf.) Raf., *Opuntia elatior* Mill., *O. dillenii,* and, mostly, *O. ficus-indica* [65,77,78,79,80,81]. Different parts of the plant (flowers, cladodes, seeds, and fruits) were explored, and different extraction methods and solvents were used according to the desired purpose. However, most of the works exploring the anti-inflammatory potential of *Opuntia* sp. were conducted using crude extracts, and only a few of them provided information on the anti-inflammatory potential of isolated phenolic molecules, especially flavonoids from *Opuntia* sp., such as isorhamnetin and kaempferol derivates [77,82,83].

### 3.1. Modulation of Inflammatory Mediators and Enzymes

Most of the available studies exploring the anti-inflammatory activity of *Opuntia* sp. extracts and isolated compounds were undertaken in vitro and explored specific mediators and enzymes involved in the inflammatory process (Table 2). Among them, nitric oxide (NO), produced by the inducible nitric oxide synthase (iNOS) upon inflammatory stimuli, is among the most explored. Gómez-Maqueo and co-workers [84] reported that the pulps and peels of two varieties of prickly pears presented anti-inflammatory potential by scavenging NO radicals. Indeed, besides being implicated in many physiological processes, the high levels of NO also play a key role in the pathogenesis of inflammation, by upregulating iNOS and pro-inflammatory cytokines’ production (TNF-α and IL-8), leading at the end to serious tissue damage [78,85]. Following this train of thought, the reduction in the inflammatory response can benefit from the well-known free radical scavenging ability of phenolic compounds. The same authors followed their study and found that the different prickly pears’ parts were also able to inhibit hyaluronidase activity. This enzyme is implicated in both physiological and pathological processes by hydrolyzing hyaluronic acid (HA), one of the most important compounds in the extracellular matrix, with more than 50% found in human skin [86]. The degradation of HA leads to the breakdown of tissues’ structural integrity and, consequently, to an increase in their permeability, favoring the progression of inflammatory mediators. Thus, a balanced regulation of hyaluronic acid metabolism is important to maintain normal tissue organization and structure of the extracellular matrix. On the other hand, an over-activation of hyaluronidase may contribute to degenerative changes in connective tissues; therefore, enzyme inhibitors may play a beneficial role during the inflammatory process, presenting potential beneficial health effects such as hindering the exacerbation of the inflammatory response [86,87,88]. Similarly, another study of Gómez-Maqueo et al. [89], using other varieties of prickly pears, explored the same bioactivities in vitro, with a notably higher efficiency for peels when compared to pulps. The authors attributed this behavior to some compounds found in the species under study, particularly to indicaxanthin and to the phenolic compounds isorhamnetin glycosides, kaempferol glycoside, and quercetin, which mainly contributed to the observed anti-inflammatory potential by presenting a higher hyaluronidase inhibitory activity when compared to other purified standards.

A study conducted by Chaalal et al. [90] demonstrated that polyphenols extracted from different parts of *O. ficus-indica* fruits (seeds, pulp, and entire fruit) presented an anti-inflammatory potential and could exert a neuroprotective effect by decreasing the transcriptional expression of pro-inflammatory mediators such as TNF-α, IL-1β, and iNOS in N13-microglial cells after lipopolysaccharide (LPS) stimulation, pointing out the potential health benefit of these compounds in case of neuronal damage. In another study, Cho et al. [78] reported that chloroform and ethyl acetate fractions of *O. humifusa* cladodes were able to decrease NO production in LPS-stimulated RAW 264.7 macrophages, pointing to quercetin as one of the main compounds responsible for this behavior. Moreover, they also noticed that both fractions differently modulated cytokines’ gene expression, especially for iNOS, IL-6, and IL-1β, suggesting the implication of other bioactive components from the species in the anti-inflammatory potential, while Yeo et al. [91] showed that methanolic extracts obtained from the seeds of the same species reduced NO production from LPS-stimulated macrophages’ RAW 264.7. The isolated compound americanin A was found to be responsible for reducing iNOS and pro-inflammatory cytokines’ (TNF-α and IL-6) expression levels, which resulted principally by preventing NF-κB translocation into the nucleus.

In a study conducted with the intestinal Caco-2/TC7 cell line, Filannino and co-workers [92] reported that raw material and fermented extracts from *O. ficus-indica* cladodes presented an anti-inflammatory effect by significantly reducing NO and chemokines’ (IL-8 and TNF-α) production, which are important effectors in the inflammatory process contributing to the recruitment and activation of different inflammatory cells. In addition to decreasing the intracellular ROS generated during cells’ stimulation, flavonoids (especially kaempferol and isorhamnetin), were considered to be the main responsible for this anti-inflammatory modulation. They displayed a significant decrease in prostaglandin E2 accumulation, which is a pro-inflammatory product from COX-2 and prostaglandin synthase metabolism, generally implicated in promoting local vasodilatation, and attraction and regulation of different immune cells’ functions. Similarly, a study conducted by Matias et al. [82] found that flavonoid-rich concentrate from *O. ficus-indica* fruits prevented oxidative stress through the neutralization of H_2_O_2_-induced free radicals and also prevented protein oxidation in inflamed Caco-2 cells. Otherwise, the extract allowed protecting the intestinal barrier dysfunction, which was correlated with the ability of some flavonoids to decrease TNF-α secretion. They also found that the incubation of inflamed Caco-2 cells with the extract significantly modulated cytokines’ secretion, leading particularly to a decrease in IL-8 and NO production, which are linked to the activation of the NF-κB pathway. Indeed, it was proven that the extract reduced the degradation of IκBα, an important inhibitor of NF-κB, preventing its migration from cytosol to the nucleus, where it could promote the transcription of pro-inflammatory genes.

### 3.2. Anti-Inflammatory Activity In Vivo

Most of the in vivo works exploring the anti-inflammatory activity of *Opuntia* species were conducted using rats as animal models, following the carrageenan-induced inflammation method, which is frequently used to evaluate the anti-edematous effect of natural products (Table 2) [79].

A study by Ahmed et al. [77] reported the in vivo anti-inflammatory potential of different parts of *O. dillenii* with a high efficiency verified for the flowers, from which three isolated compounds, namely, kaempferol 3-*O*-α-arabinoside, isorhamnetin-3-*O*-β-d-glucopyranoside, and isorhamnetin-3-*O*-β-d-rutinoside, were characterized as the active principles contributing significantly to reducing paw edema in albino rats. Ammar et al. [65] also reported the anti-inflammatory potential for *Opuntia* sp. flowers; the authors found that the methanolic extracts of *O. ficus-indica* flowers exhibited an anti-inflammatory potential by reducing the paw edema size in Wistar rats, with the same efficiency as the non-steroidal anti-inflammatory drug indomethacin. This effect was confirmed by a significant decrease in the number of inflammatory cells, including leukocytes and lymphocytes, a decrease in malondialdehyde (MDA) levels, which is correlated with the decrease in the lipid peroxidation process occurring at inflammatory sites, and a restoration of some antioxidant enzymes’ activities, including superoxide dismutase (SOD), catalase (CAT), and glutathione (GSH), which contribute to neutralize free radicals’ overproduction. According to the phytochemical analysis, the authors suggested the implication of phenolic compounds, particularly quercetin, isorhamnetin, and kaempferol, which could scavenge free radicals and decrease inflammation. Moreover, Antunes-Ricardo et al. [83] found that *O. ficus-indica* cladodes’ extract and its isolated isorhamnetin derivatives (isorhamnetin-3-*O*-glucosyl-rhamnosyl-rhamnoside and isorhamnetin-3-*O*-glucosyl-rhamnoside) decreased the amount of neutrophils’ infiltration into the inflammatory site (carrageenan-induced air-pouch inflammation in rats) with a decrease in NO production more efficient than that obtained with the standard drug used, indomethacin. The authors also found that cladodes’ extract and isolated compounds were able to inhibit COX-2 activity and cytokines’ production, particularly TNF-α and IL-6, with a better efficiency for the crude sample, probably due to a synergistic effect of its different phytochemicals.

## 4. Conclusions

The present review shows the richness of *Opuntia* species as producers of a wide variety of phenolic compounds, with an important role in the inflammatory process. The anti-inflammatory studies conducted until now demonstrated the benefit of different species to reduce the oxidative stress occurring at the site of an injury, decreasing the amount of neutrophils’ infiltration, as well as pro-inflammatory mediators’ production, such as NO, TNF-α, and interleukins. These effects were correlated with the presence of flavonoids in the different tissues of cactus, namely, quercetin, isorhamnetin, and kaempferol derivates, as the important bioactive components. Even though the *Opuntia* genus regroups a lot of species throughout the world, the available studies are limited to a few of them, with *O. ficus-indica* being by far the most explored. Nevertheless, reports on cactus plants reveal their potential anti-inflammatory application in the pharmaceutical industry, supporting the traditional use of these species in folk medicine and enhancing their economic value worldwide and for local communities. 

## Figures and Tables

**Figure 1 molecules-27-04763-f001:**
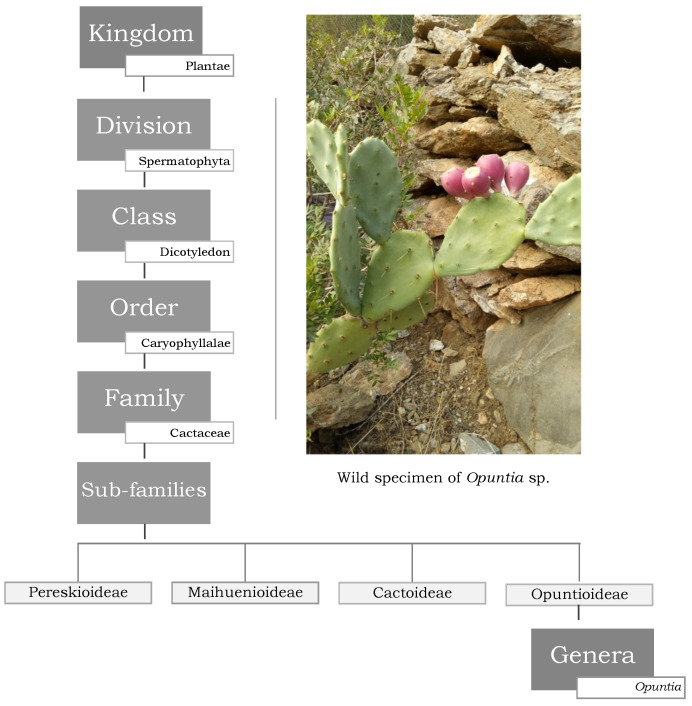
Taxonomic classification of *Opuntia* species (photograph from the author (W.Z.): wild *Opuntia* sp., Bejaia, Algeria).

**Figure 2 molecules-27-04763-f002:**
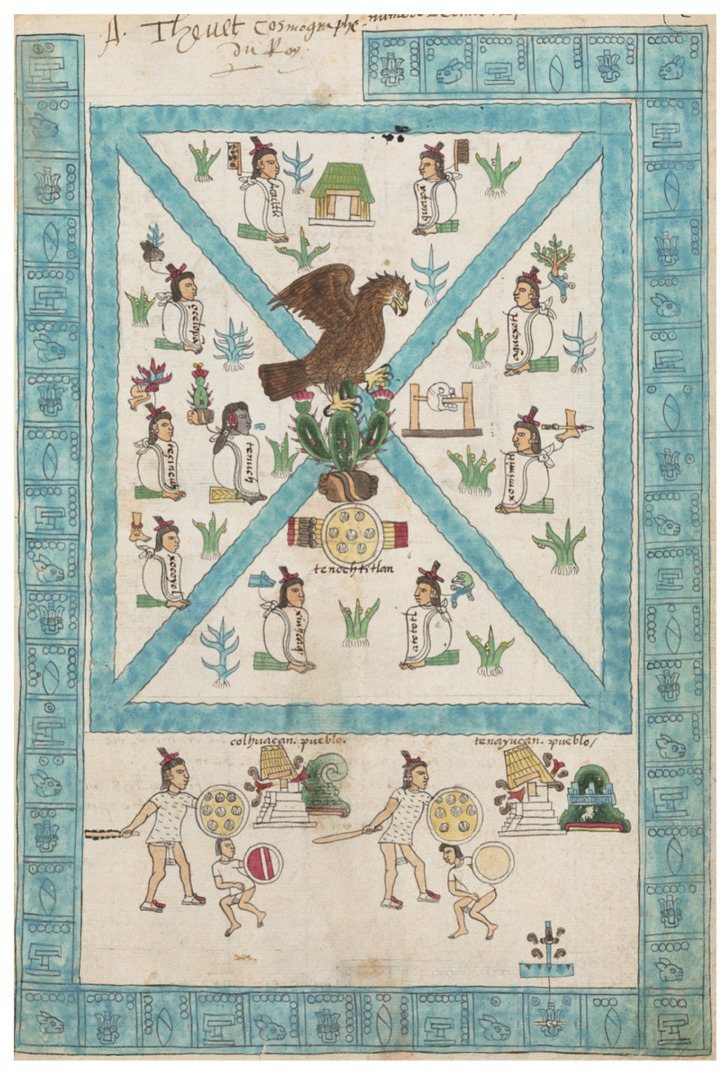
Representation of the emblem of Tenochtitlan from the Codex Mendoza, with the prickly pear as the center of the universe. Photo: © Bodleian Libraries, University of Oxford; Shelfmark: Bodleian Library MS. Arch. Selden. A. 1; Holding Institution: Bodleian Libraries, University of Oxford; Terms of use: CC-BY-NC 4.0.

**Figure 3 molecules-27-04763-f003:**
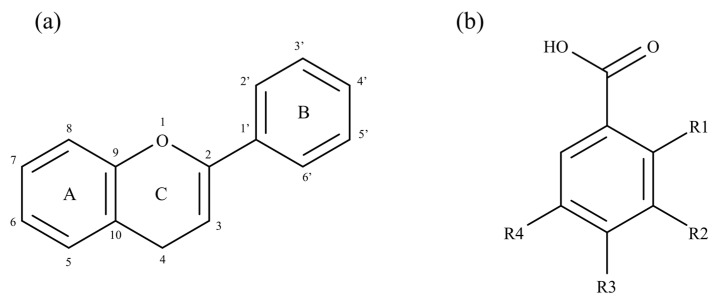
Basic structures of flavonoids (**a**) and phenolic acids (**b**).

**Figure 4 molecules-27-04763-f004:**
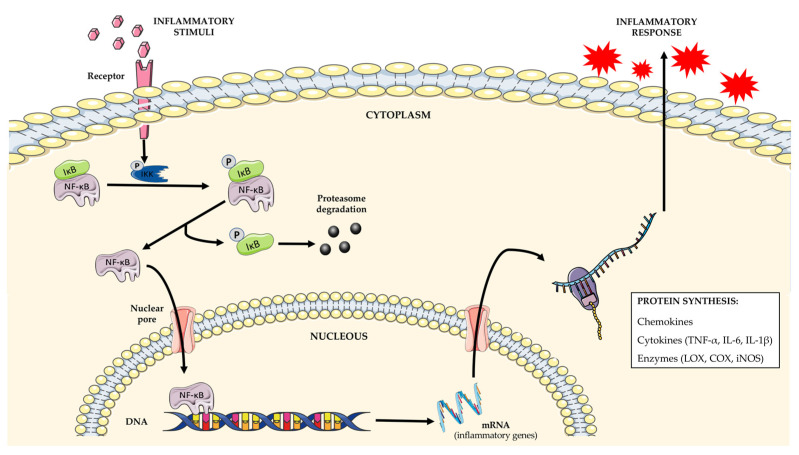
The implication of NF-κB signaling pathway in inflammation. The inflammatory stimulation of a cell may lead to IKK phosphorylation and activation which, in turn, may lead to the phosphorylation of the IκB/NF-κB complex retained in the cytoplasm. The IκB is degraded by the proteasome, while the NF-κB transcription factor can enter into the nucleus and bind DNA to initiate the transcription of some targeted genes implicated in the inflammatory response.

**Table 1 molecules-27-04763-t001:** Phenolic compounds reported in the different vegetative parts of the genus *Opuntia*
^1^.

Phenolic Compounds	Plant Tissue	Concentration (µg/g)	*Opuntia* Species	References
**Flavonoids**				
*Flavones*				
Apigenin	Seeds Cladodes Flowers	NS 0.19–0.65 NS	*O. stricta* *O. ficus-indica* *O. hyptiacantha* *O. streptacantha* *O. megacantha * *O. albicarpa*	[38,42,43,44]
Luteolin	Pulp Peel	NS NS	*O. ficus-barbarica* *O. robusta*	[45]
*Flavonols*				
Myricetin	Seeds Pulp Peel Cladodes	198.19–428.14 NS NS 8.52	*O. ficus-indica* *O. ficus-barbarica * *O. robusta*	[45,46,47]
Rutin	Seeds Pulp Peel Cladodes	8.00–100.00 9.70–12.50 65.70–103.40 2.11–4.95	*O. ficus-indica* *O. ficus-barbarica* *O. hyptiacantha* *O. streptacantha* *O. megacantha * *O. albicarpa*	[42,44,45,48,49]
Quercetin and derivates	Seeds Pulp Peel Cladodes Flowers	4.37–18.77 84.20–599.20 715.70–1316.20 8.97–75.13 NS	*O. ficus-indica* *O. ficus-barbarica * *O. robusta * *O. engelmannii* *O. streptacantha * *O. hyptiacantha * *O. megacantha* *O. albicarpa*	[12,38,42,44,45,47,49,50,51,52,53,54,55,56,57,58]
Kaempferol and derivates	Pulp Peel Cladodes Flowers	207.10–529.10 52.90–675.50 72.97–241.68 321.00–708.00	*O. ficus-indica* *O. engelmannii* *O. streptacantha * *O. hyptiacantha* *O. megacantha* *O. albicarpa* *O. microdasys*	[38,47,48,49,50,52,53,54,55,57,59]
Isorhamnetin and derivates	Seeds Pulp Peel Cladodes Flowers	67.14–288.58 29.30–58.40 1484.70–2213.70 1250.00–4140.00 NS	*O. ficus-indica* *O. microdasys* *O. stricta * *O. streptacantha * *O. hyptiacantha * *O. megacantha* *O. albicarpa*	[12,19,38,44,46,47,48,50,51,52,53,54,55,56,57,59,60]
*Flavanones*				
Naringenin	Pulp Peel	210.00 20.00–180.00	*O. ficus-indica* *O. ficus-barbarica* *O. robusta*	[45,56]
*Flavanols*				
Catechin	Seeds Pulp Peel Cladodes Flowers	NS 14.44–27.89 NS 180.00 NS	*O. stricta* *O. ficus-indica* *O. megacantha* *O. streptacantha* *O. robusta*	[17,38,43,49,52,54,61]
Epicatechin	Seeds Pulp Peel	NS 19.16–90.81 NS	*O. ficus-indica* *O. albicarpa* *O. megacantha* *O. streptacantha* *O. robusta*	[17,42,61]
Gallocatechin	Seeds Pulp Peel	NS 116.60–178.20 120.40–334.70	*O. stricta* *O. ficus-indica*	[43,49]
Epigallocatechin	Seeds	NS	*O. stricta* *O. ficus-indica*	[42,43]
*Anthocyanidins*				
Pelargonidin	Seeds Cladodes	NS 187.97	*O. stricta* *O. ficus-indica*	[43,47]
Cyanidin	Seeds Cladodes	NS 1058.57	*O. stricta* *O. ficus-indica*	[43,47]
Delphinidin	Cladodes	2.81	*O. ficus-indica*	[47]
Petunidin	Cladodes	186.55	*O. ficus-indica*	[47]
Malvidin	Cladodes	4.31	*O. ficus-indica*	[47]
**Phenolic Acids**				
Gallic acid and derivates	Seeds Pulp Peel Cladodes Flowers	NS 32.60–81.20 NS 20.53–38.96 NS	*O. ficus-indica* *O. stricta* *O. ficus-barbarica * *O. robusta * *O. albicarpa* *O. megacantha* *O. streptacantha* *O. hyptiacantha*	[17,42,43,44,45,49,54,61,62,63]
Ferulic acid and derivates	Seeds Pulp Peel Cladodes Flowers	96.33–1366.24 80.00 150.00–390.00 130.00–370.00 291.00–786.00	*O. ficus-indica* *O. stricta* *O. ficus-barbarica * *O. microdasys* *O. hyptiacantha* *O. streptacantha* *O. megacantha* *O. albicarpa*	[12,43,44,45,46,48,49,51,52,55,56,59,61,64]
Caffeic acid and derivates	Seeds Pulp Peels Cladodes Flowers	NS NS NS NS 255.00–469.00	*O. ficus-indica* *O. ficus-barbarica* *O. robusta * *O. microdasys* *O. hyptiacantha* *O. streptacantha* *O. megacantha * *O. albicarpa*	[12,38,42,44,45,48,49,51,54,59,61]
Sinapic acid	Seeds Pulp Peel Cladodes	NS 100.00–4100.00 820.00–2350.00 40.00–750.00	*O. stricta* *O. ficus-indica*	[43,49,56]
*p*-Coumaric acid	Seeds Pulp Peel Cladodes Flowers	NS NS NS 20.91 65.00–178.00	*O. ficus-indica* *O. ficus-barbarica * *O. robusta * *O. microdasys* *O. hyptiacantha* *O. streptacantha* *O. megacantha * *O. albicarpa*	[42,44,45,48,49,52,59]
Hydroxycinnamic acid	Cladodes	8.45–1248.24	*O. ficus-indica* *O. hyptiacantha* *O. streptacantha* *O. megacantha* *O. albicarpa*	[44,47]
Chlorogenic acid	Seeds Cladodes	885.31–1148.41 5.00–26.49	*O. ficus-indica* *O. streptacantha * *O. hyptiacantha* *O. megacantha * *O. albicarpa*	[42,44,46,50,52]
Ellagic acid	Seeds Pulp Peel	73.74–74.38 25.00–73.20 NS	*O. ficus-indica* *O. megacantha* *O. streptacantha* *O. robusta* *O. ficus-indica*	[17,46,61]
Vanillic acid	Seeds Pulp Peel Cladodes Flowers	NS NS NS 0.11–24.30 NS	*O. stricta* *O. ficus-barbarica* *O. robusta * *O. ficus-indica* *O. hyptiacantha* *O. streptacantha* *O. megacantha * *O. albicarpa*	[43,44,45,49,54,61]
Syringic acid	Seeds Pulp Peel Cladodes Flowers	NS 13.60–66.50 NS 2.34–13.99 NS	*O. ficus-indica* *O. robusta * *O. albicarpa* *O. megacantha* *O. streptacantha* *O. hyptiacantha* *O. stricta*	[12,17,42,43,44,45,54,61]
Protocatechuic acid	Seeds Pulp Peel	4.57–22.36 NS NS	*O. ficus-indica* *O. ficus-barbarica * *O. robusta* *O. stricta*	[45,46,49,58,61,62,63]
Hydroxybenzoic acid	Pulp Peel Cladodes	200.90–816.80 964.00–1718.20 114.01	*O. ficus-indica* *O. hyptiacantha* *O. streptacantha* *O. megacantha * *O. albicarpa*	[44,47,49,57,62]
Piscidic acid	Seeds Pulp Peel Cladodes	NS NS NS NS	*O. ficus-indica* *O. stricta*	[48,51,53,55,57,58,63]
Eucomic acid	Seeds Pulp Peel Cladodes	NS NS NS NS	*O. ficus-indica* *O. streptacantha * *O. hyptiacantha * *O. megacantha * *O. albicarpa * *O. stricta*	[48,50,51,53,55,58,63]
Gentisic acid	Pulp Peel	NS NS	*O. ficus-barbarica * *O. robusta*	[45]
Rosmarinic acid	Peel Flowers	NS NS	*O. ficus-indica*	[49,54]
Catechol	Seeds Pulp Peel	NS NS NS	*O. stricta* *O. ficus-barbarica * *O. robusta * *O. ficus-indica*	[43,45,61]
**Other Phenolics**				
Phloretin Psoralen Pinoresinol	Seeds	NS	*O. stricta*	[43]

^1^ NS, not specified.

**Table 2 molecules-27-04763-t002:** Anti-inflammatory potential and mechanism of action of phenolic compounds extracted from *Opuntia* species ^1^.

	Species (Tissue)	Compounds	Dose	Model	Mechanism of Action	Ref.
** In Vitro Studies **	* Opuntia ficus-indica* (seeds, pulp, fruits)	Phenolic compounds from crude extracts	10 mg/mL	LPS-stimulated murine N13 microglial cells	- Downregulation of TNF-α, IL-1β, and iNOS expression	[90]
	* Opuntia humifusa* (cladodes)	Phenolic compound from crude extracts	0.05/0.1 mg/mL	LPS-stimulated Macrophages’ RAW 264.7	- ↓ NO production - Downregulation of iNOS, IL-1β, and IL-6 genes’ expression	[78]
	* Opuntia humifusa* (seeds)	Isoamericanin A	1.0–4.0 µg/mL	LPS-stimulated Macrophages’ RAW 264.7	- ↓ TNF-α, IL-6, and iNOS expression levels - ↓ NF-Κb levels in the nucleus by inhibition of IκB phosphorylation	[91]
	* Opuntia ficus-indica* (cladodes)	Crude extract	10 mg/mL	Human intestinal Caco-2/TC7 cells	- ↓ NO, TNF-α, and IL-8 production - Intracellular reduction in reactive species - ↓ Prostaglandin synthesis	[92]
	* Opuntia ficus-indica* (fruits)	Polyphenols from crude extract	0.05 mg/mL	Human colon carcinoma Caco-2 cells	- ↓ H_2_O_2_-induced reactive species - Prevention of H_2_O_2_–induced protein oxidation - Cell protection from barrier dysfunction - ↓ NO, TNF-α, and IL-8 secretion - ↓ IκBα depletion	[82]
** In Vivo Studies **	* Opuntia dillenii* (stems, flowers, fruits)	Kaempferol 3-*O*-α-arabinoside Isorhamnetin-3-*O*-β-d-glucopyranoside Isorhamnetin-3-*O*-β-d-rutinoside	50 mg/kg BW	Carrageenan-induced paw edema in Albino rats	- ↓ Edema formation	[77]
	* Opuntia ficus-indica* (flowers)	Phenolic compounds from crude extract	400 mg/kg BW	Carrageenan-induced paw edema in Wistar rats	- ↓ Edema formation - ↓ Amount of immune cells - Neutralization of lipid peroxidation induced by reactive species - ↑ CAT, SOD, and GSH activities	[65]
	* Opuntia ficus-indica* (cladodes)	Isorhamnetin-3-*O*- glucosyl-rhamnoside Isorhamnetin-3-*O*- glucosyl-rhamnosyl-rhamnoside	5 mg/kg BW	Carrageenan-induced air-pouch inflammation in Wistar rats	- ↓ Edema formation - ↓ Total leukocytes’ amount - Inhibition of COX-2 activity - ↓ NO, TNF-α, and IL-6 production	[83]

^1^ BW, body weight; CAT, catalase; COX, cyclooxygenase; GSH, glutathione; H_2_O_2_, hydrogen peroxide; IL, interleukin; iNOS, inducible nitric oxide synthase; LPS, lipopolysaccharide; NO, nitric oxide; SOD, superoxide dismutase; TNF, tumor necrosis factor.

## Data Availability

Not applicable.

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
