# Peer review of "Phenolic Compounds’ Occurrence in *Opuntia* Species and Their Role in the Inflammatory Process: A Review"

_molecules, 2022, doi:10.3390/molecules27154763_

Round 1

Reviewer 1 Report

This review article is very interesting, novel in terms of the visibility that the manuscript offers to know the plants of the Opuntia genus and to evidence from cellular and molecular biology the anti-inflammatory property observed with traditional medicine. However there are some details that can improve the manuscript. These aspects are:

1. page 3, line 64: the authors refer to the number of polyphenols identified according to the reference, 8000 compounds. However, in line 115 they mention that the 8000 compounds correspond to phenolic compounds, which in turn include both phenolic acids and polyphenols. Correct for consistency.

Table 1: I would recommend including a column indicating the concentration of the compounds mentioned in the tissues indicated in the table. 

3. Line 162: The authors mention that it is difficult to establish the phenolic compounds profile of Opuntia species plants, I suggest mentioning what differences there are between the methodologies used for their analysis in the same tissue.

4. Figure 4: please indicate if it is original of the authors or adapted from another reference.

5. In Table 2: I suggest including a column indicating dose and exposure to treatment.

6. Reference table 2 in section 3.2.

7. In section 4: challenges and perspectives, it looks like a conclusion section and does not mention the challenges of science in terms of obtaining a phenolic profile of this plant or some of its tissues, diseases with inflammatory states of potential application according to existing preclinical evidence, etc.

Author Response

Answer to Reviewer 1

First we would like to thank the reviewer for his overall appreciation of our manuscript, and for his timely comments and suggestions which contributed to improve the overall quality and clarity of our manuscript.

This review article is very interesting, novel in terms of the visibility that the manuscript offers to know the plants of the Opuntia genus and to evidence from cellular and molecular biology the anti-inflammatory property observed with traditional medicine. However there are some details that can improve the manuscript.

We thank the reviewer for his comment. We believe this review can be a valuable tool for those working in the exploitation of Opuntia phenolic profile and anti-inflammatory potential, enhancing the economic and biotechnological value of Opuntia species.

These aspects are:

  1. page 3, line 64: the authors refer to the number of polyphenols identified according to the reference, 8000 compounds. However, in line 115 they mention that the 8000 compounds correspond to phenolic compounds, which in turn include both phenolic acids and polyphenols. Correct for consistency.

We thank the reviewer comment and apologize for the inconsistency. The text was reviewed and corrected as suggested. Please see Line 64 on page 3 and of the revised version of the manuscript.

Table 1: I would recommend including a column indicating the concentration of the compounds mentioned in the tissues indicated in the table.

We thank the reviewer for his valuable suggestion. Although the concentration of the identified compounds is not always mentioned, and most of the cited works refer only to qualitative analysis, all the references were carefully checked by us and, whenever possible, the concentration of the phenolic compounds was included in a new column (Dose), in Table 1, as suggested. Every time the information was not available, we mentioned “NS” in the table, meaning “Not Specified” Please see the updated Table 1, in the revised version of our manuscript.

  1. Line 162: The authors mention that it is difficult to establish the phenolic compounds profile of Opuntia species plants, I suggest mentioning what differences there are between the methodologies used for their analysis in the same tissue.

We apologize the reviewer for the imprecision. The text was modified for better clarification of our statement. Please see Lines 165-171, on pages 8 and 9 of the revised version of our manuscript.

  1. Figure 4: please indicate if it is original of the authors or adapted from another reference.

We apologize for the lack of this information. Figure 4 was entirely drawn by the authors. This information was added in figure legend, in the revised version of the manuscript.

  1. In Table 2: I suggest including a column indicating dose and exposure to treatment.

As for Table 1, we have checked all the references cited in the table, and added the suggested information in a new column. Please see the updated Table 2, in the revised version of our manuscript.

  1. Reference table 2 in section 3.2.

We apologize for the lapse. Table 2 was referred in section 3.2 as suggested.

  1. In section 4: challenges and perspectives, it looks like a conclusion section and does not mention the challenges of science in terms of obtaining a phenolic profile of this plant or some of its tissues, diseases with inflammatory states of potential application according to existing preclinical evidence, etc.

We agree with the reviewer. The Conclusion section was updated accordingly. Please see the Conclusion of the revised manuscript.

Reviewer 2 Report

The Review about the polyphenolic composition of Opuntia species and their anti-inflammatorry role is well written.

Only minor changes are needed.

In details:

-line 16: "comprises" and not "comprise"

-line 21: I suggest antimicrobial instead of "anti-microbial"

- line 73: I suggest to stop the sentence after "system", because it is too long.

- Section 2: I suggest

2.Phenlic compounds

2.1. General overview

2.2. Phenolic compounds....

Moreover, when possible avoid to repeat Opuntia and use the abbreviation O. (lines 165, 166, 176, 236,322...)

Table 2 should be formatted according to the Instruction for Authors

Add reference to Table 2 in the text in the section 3.2

Author Response

Answer to Reviewer 2

First we would like to thank the reviewer for the time spent analysing our manuscript, and for his suggestions that contributed to the improvement of our work.

The Review about the polyphenolic composition of Opuntia species and their anti-inflammatory role is well written.

Only minor changes are needed.

We thank the reviewer for his positive appreciation of our work.

In details:

-line 16: "comprises" and not "comprise"

Correction has been made as suggested. Please see line 16 on page 1 of the revised manuscript.

-line 21: I suggest antimicrobial instead of "anti-microbial"

The correction has been made as suggested. Please see line 21 on page 1 of the revised manuscript.

- line 73: I suggest to stop the sentence after "system", because it is too long.

We thank the reviewer for his suggestion. The sentence was re-written and divided as suggested. Please see lines 72 to 76 on page 3 of the revised manuscript.

- Section 2: I suggest

2.Phenolic compounds

2.1. General overview

2.2. Phenolic compounds....

We agree with the reviewer. Sections numbering was updated as suggested. Please see the revised manuscript.

Moreover, when possible avoid to repeat Opuntia and use the abbreviation O. (lines 165, 166, 176, 236,322...)

We apologize for the inconsistency. The whole manuscript was carefully checked and abbreviations were used whenever applicable. In order to follow the correct botanic description, we had included the name of the botanic descriptor of the species, when the species name is refereed for the first time, for instance: Opuntia ficus indica L (Mill) and then O. ficus indica; Opuntia microdasys (Lehm.) N.E. Pfeiffer and then O. microdasys, and so on. Please see the revised version of the manuscript.

Table 2 should be formatted according to the Instruction for Authors

We thank the reviewer for his comment. We have checked the formatting of Table 2 and add the updated version in the revised manuscript.

Add reference to Table 2 in the text in the section 3.2

We apologize for the lapse. Table 2 was referred in section 3.2 as suggested. Please see section 3.2 of the revised manuscript.

Reviewer 3 Report

The present review is interesting and methodologically correct. The references used are appropriate and allow the authors to clearly highlight the anti-inflammatory potential of the Opuntia species as well as the role that phenolic compounds play in its bioactivity.

Author Response

Answer to Reviewer 3

The present review is interesting and methodologically correct. The references used are appropriate and allow the authors to clearly highlight the anti-inflammatory potential of the Opuntia species as well as the role that phenolic compounds play in its bioactivity.

We are very grateful to the reviewer for his positive appreciation of our work. We hope our manuscript can constitute an important reference for the research groups working in the chemical characterization of Opuntia sp. phenolic compounds, as well as to valorize Opuntia species from the economic and biotechnological point of view.